# Open Campus Policies: How Built, Food, Social, and Organizational Environments Matter for Oregon’s Public High School Students’ Health

**DOI:** 10.3390/ijerph17020469

**Published:** 2020-01-10

**Authors:** Elizabeth L. Budd, Raoul S. Liévanos, Brigette Amidon

**Affiliations:** 1Department of Counseling Psychology and Human Services, College of Education, University of Oregon, 5461 University of Oregon, Eugene, OR 97403, USA; 2Department of Sociology, College of Arts and Sciences, University of Oregon, 1291 University of Oregon, Eugene, OR 97403, USA; raoull@uoregon.edu; 3Morgridge College of Education, University of Denver, 1999 E. Evans Avenue, Denver, CO 80208, USA; Brigette.Amidon@du.edu

**Keywords:** schools, youth, policy, built environment, food environment, social environment, organizational environment

## Abstract

Open campus policies that grant access to the off-campus food environment may influence U.S. high school students’ exposure to unhealthy foods, yet predictors of these policies are unknown. Policy holding and built (walkability), food (access to grocery stores), social (school-to-neighborhood demographic similarity), and organizational (policy holding of neighboring schools) environment data were collected for 200 Oregon public high schools. These existing data were derived from the Oregon School Board Association, WalkScore.com, the 2010 Decennial Census, the 2010–2014 American Community Survey, the Supplemental Nutrition Assistance Program, TDLinex, Nielson directories, the U.S. Department of Education, the National Center for Education Statistics, and the Common Core of Data. Most (67%) of Oregon public high schools have open campus policies. Logistic regression analyses modeled open campus policy holding as a function of built, food, social, and organizational environment influences. With health and policy implications, the results indicate that the schools’ walkability, food access, and extent of neighboring open campus policy-schools are significantly associated with open campus policy holding in Oregon.

## 1. Introduction

### 1.1. Background

Among U.S. youth, frequent consumption of fast food is positively associated with less healthy diets (e.g., higher total caloric intake, more servings of sugar-sweetened beverages, fewer servings of fruits and vegetables) [1]. An extensive body of research attributes youth dietary behavior to varied cognitive, affective, habitual, and environmental influences [2]. Empirical research is increasingly concerned with assessing the environmental influences on youth’s diets [2], reinforcing the notion that environmental-level interventions hold the most promise for producing population-level changes [3]. Systematic reviews [4,5] have indicated that future research should pay better attention to schools and their food policies that can affect youth dietary behaviors. For example, the mere number of on- and off-school campus food options to which high school students have access is positively associated with the intake of sugar-sweetened beverages, high-fat foods, and fast food meals [6]. In California, USA, a convenience store presence within a 10-min walk from schools has been positively associated with overweight status among ninth-grade public high school students [7].

These general problems of youth’s health and environment unfold unevenly across U.S. sub-populations. Research finds that U.S. neighborhoods that are rural, of low socioeconomic status (SES), and/or have primarily racial minority residents are more likely to experience limited access to grocery stores and healthy food options [8]. Meanwhile, fast food restaurants and/or convenience stores with unhealthy food options are more prevalent in low SES and racial minority neighborhoods and around low SES schools [8,9,10,11,12]. These conditions can amplify the risk of adopting unhealthy eating behaviors among youth and students from low SES and racial minority backgrounds, contributing to population health disparities.

Open campus policies in U.S. high schools (grades 9–12) grant access to the off-school campus food environment. As such, they play an important role in increasing or decreasing the exposure of the country’s most vulnerable students to the unhealthy food environments that surround their schools. For instance, one study found that high school students were more likely to eat lunch at fast food restaurants if their high school had an open campus policy, as opposed to a closed campus policy that restricts access to off-campus food [13].

Additional research on schools with open campus policies is limited. Among this research, one study found that high schools with open campus policies had higher rates of motor vehicle accidents among high school students and lower participation in the National School Lunch Program (NSLP) compared to schools with closed campus policies [14]. In a study of U.S. schools across grade levels, approximately 40% of students eligible for the NSLP participate, and rates of NSLP participation among high school students average about half of that of elementary school students [15]. Regarding these high school students who qualify for NSLP, studies show that they eat more nutritiously when they eat NSLP lunches rather than seek food elsewhere [16]. However, if these students have easy access to fast food options, then they are more likely to engage in those less healthy options [16].

The impact of various school policies (e.g., relating to school wellness or physical education) on behavioral and health outcomes of students is well documented. However, open campus policies have gone largely unmentioned in previous research [17,18,19]. For example, none of the commonly used local school wellness policy evaluation tools (e.g., Colorado Smart Source, New Jersey Wellness Policy Assessment Tool, WellSAT 3.0) assess open campus policy holding [20,21,22]. Likewise, a systematic review found that among 40 studies on fast food access surrounding schools, none of them directly examined open campus policies or predictors of a school’s open versus closed campus policy holding [10].

### 1.2. The Present Study

This study begins to fill existing research gaps on open campus policies. Using a case study of public high schools in Oregon, USA, we describe the prevalence of open campus policies and identify multiple environmental predictors of open campus policies among those schools. We selected the state of Oregon as a case study due to the following considerations. First, the prevalence and predictors of open campus policies in the state are unknown. Second, 35% of residents live in rural communities and 47% of students are eligible for the NSLP, and both of these conditions are risk factors for childhood obesity [23,24]. Without an Oregon-specific understanding of the environmental predictors of open campus policies, school policy-makers may be operating in the dark regarding a policy decision that could impact the health and safety of students, particularly those most vulnerable to health disparities.

Our empirical analysis of open campus policy holding among Oregon public high schools is guided by theoretical considerations pertaining to environmental and contextual factors that influence organizational practice. We draw from strands of ecological, systems, organizational, and institutional theories regarding the contextual and material conditions that influence organizations [25,26,27,28,29]. We account for these conditions by exploring the extent to which the built environment (the degree of pedestrian travel accessibility) and the food environment (the degree of access to grocery stores with healthy food options) are predictors of open campus policy holding.

We conceptualize the social environment of the schools as another influential condition that affects open campus policy holding. In so doing, we draw on the broader sociological literature related to social homophily in shaping social and spatial relations [30,31,32,33,34,35]. Social homophily refers to “the preference of individuals to associate with others who are similar to them” [32] (p. 397), which has consequences for the degree to which socially heterogeneous individuals and groups interact in geographic space [30,31,32,33,34,35].

We consider homophily in race and class terms due to the ongoing legacy and influence of race and class on U.S. school and residential segregation. Despite the passage of significant U.S. civil rights-era federal and local desegregation policies from the 1950s onward, various school desegregation efforts, particularly “busing”, did not “fully desegregate public schools because school officials, politicians, courts, and the news media valued the desires of white parents more than the rights of black students” [36] (p. 2). Recently, research suggests that the ongoing flight of white and upper-income families from public schools has contributed to elevated “percentages of poor children in neighborhood schools…than in their corresponding catchment areas and this difference is greater when the majority of children living in a neighborhood are racial minorities” [37] (p. 1227). Such dynamics are manifest in Oregon, whose state charter and governing authorities supported land acquisition primarily for white settlers. In Portland, the largest metropolitan area of Oregon, racist real estate practices and state and school policies created large racial and class inequalities in the distribution of school and neighborhood amenities [38,39,40]. Drawing on this literature, we explore the extent to which school-to-neighborhood SES and racial similarity is a predictor of an Oregon public high school holding an open campus policy, which would permit students to leave campus and interact with the proximate SES and racial conditions.

Lastly, we consider the organizational environment in which Oregon public high schools operate. Organizational environments are constituted by formal rules and procedures, normative conventions, and taken-for-granted models for dealing with uncertainty [29,41,42,43,44,45]. This context can compel organizations to succumb to isomorphic pressures to conform to dominant structures and practices in their organizational environment in ways that influence how organizations operate and hold influence over social relations in various spatial and temporal contexts [27,28,46,47]. We consider the organizational environment of Oregon public high schools by accounting for the prevalence of open campus policies of neighboring public high schools in a manner that also addresses concerns about spatial dependence in our regression analysis.

## 2. Materials and Methods

### 2.1. Unit of Analysis and Dependent Variable

We collected data on open campus policy holding during the fall of 2017 for all public high schools (not including charter or alternative schools) in the state of Oregon (*n* = 200) using the Oregon Department of Education’s “Oregon School Directory” [48]. We determined the policy holding status for each high school using the following techniques. We began by reviewing school district policies provided on the Oregon School Board Association’s website [49], individual school districts’ online policy manuals, and individual high school’s student handbooks. In a handful of cases, we determined open campus policy holding by reaching out to high school principals via email. Campus policies were originally organized into three categories: closed campus (students are not allowed to leave campus at any time during the school day); open campus (students are allowed to leave campus at lunch, and during periods they do not have class); and restricted open campus (some students are permitted to leave at lunch or the school has additional provisions in place to supervise and regulate open lunch periods). For the purposes of these analyses, open campus and restricted open campus policies were collapsed into open campus policy holding, consistent with other studies [14]. From this, we derived our dichotomous dependent variable: *whether a school has an open campus policy* (1 = yes; 0 = no). Data are available via Dataverse [50].

### 2.2. Independent Variables

#### 2.2.1. Built Environment: Walkability

We operationalized the built environment with *walkability scores* for each school. We gathered these scores by entering the physical address of each school into the Walk Score website (https://www.walkscore.com/) [51]. This website analyzes walking routes from a given address to surrounding amenities (e.g., stores and restaurants) [52]. Walk Scores are widely used in research across disciplines and have been shown to produce valid measures of walkability in the United States and other countries [53,54,55,56]. Walk Scores feature a point system, where points are assigned based on proximity to amenities. Maximum points are given to amenities located within a 5-min walk Euclidean radius (0.25 miles or 402.37 m), and fewer points are given for more distant amenities. Amenities beyond a 30-min walk receive no points. Walk Scores also account for how supportive the environment is for pedestrians by including population density and various road metrics (e.g., block length and intersection density) in the calculations. Addresses located in areas that are more pedestrian-friendly receive higher walk scores. Scores of 0–24 indicate “almost all errands require a car”, while scores of 25–49 mean “most errands require a car”, 50–69 mean “some errands can be accomplished on foot”, 70–89 mean “most errands can be accomplished on foot”, and 90–100 mean “daily errands do not require a car” [51].

#### 2.2.2. Food Environment: Food Desert Exposure

Food deserts refer to the extent to which residents and/or schools lack access to a supermarket or have limited access to healthy and affordable food in their “local food environment” [57]. At the time of this study, we did not have access to the geographic locations of establishments typically used in previous food desert research as markers of healthy or unhealthy off-campus food options [52]. Instead, we drew on publicly available results of a U.S. Department of Agriculture-sponsored study to assess school exposure to food desert census tracts within a 500-m Euclidean buffer around the schools [58]. We selected the 500-m threshold to approximate a pedestrian environment [59] and to maintain consistency with the 5-min walk radius used to construct our built environment walkability indicator described above.

We used Rhone et al.’s [58] different classifications of “low access” food deserts instead of their classification of “low-income” food deserts in our analysis of Oregon public high schools’ food environments. We based this decision on our objective to develop a parsimonious regression model wherein we use alternative and related measures of the social environment discussed below (see Section 2.2.3). The low access measures derived from the 2010 Decennial Census, the 2010–2014 American Community Survey, locational data from the Supplemental Nutrition Assistance Program and TDLinex, and the Nielson directories of supermarkets and large grocery stores that serve as “proxies for the complete set of stores that sell a wide variety of healthy foods at affordable prices” [58] (p. 2). Three indicators of low access tracts that we included are Euclidean distance-based measures that assess the extent to which a census tract centroid is one mile (1.61 km) from the nearest store in urban areas (i.e., places with more than 2500 people) or 10 or 20 miles (16.09 or 32.19 km, respectively) in rural areas (i.e., places with less than 2500 people). The fourth low access measure considers the transit-dependence of tracts. Rhone et al. [58] classified transit-dependent tracts as low access food deserts if at least 100 of the total number of households did not have a vehicle, and the tract was beyond 0.5 miles (804.67 m) from a supermarket. Low access transit-dependent food desert tracts were also classified as such if greater than or equal to 500 individuals are beyond 20 miles (32.19 km) from a supermarket; or greater than or equal to 33% of individuals are beyond 20 miles from a supermarket [58].

We accounted for the extent to which schools were exposed to those four different classifications of low-access food desert tracts. We did this by calculating the total area of the school 500-m buffer that was covered by the different low access food desert tracts. We interpret these variables as indicating higher exposure to food deserts when greater shares of a school’s buffer are covered by low access food desert tracts. Figure 1 illustrates how we constructed the four food environment variables.

The 500-m buffers for 82 (41%) of the 200 schools intersected 1-mile/1.61-km food desert tracts. Of those 82 high schools, 44 (53.66%) were exposed to only those types of food desert tracts. The 500-m buffers for 24 (54.54%) of these 44 high schools were completely covered by 1-mile/1.61 km food desert tracts. Figure 1, Map B, uses the example of how 100% of the 500-m buffer for the open campus, Tualatin High School, intersected four different 1-mile/1.61-km food desert tracts. Analogous to similar distance-based exposure methods using buffers [60], Figure 1B also shows how our approach to measuring food desert exposure permits us to account for schools’ location in food desert tracts and schools’ spatial proximity to neighboring food desert tracts within the pedestrian environment of the schools.

We found in our exposure assessment that there were eight instances of triple exposure of schools to low-access food desert tracts. Maps C–E in Figure 1 use the case of the rural, open-campus, Bonanza Junior/Senior High School of Klamath County to illustrate this triple exposure and how it can vary for one school. Figure 1C shows how this school’s buffer was completely covered by two different 10-mile/16.09-km food desert tracts. Figure 1D,E show how Bonanza’s 500-m buffer only intersected 16.48% of the boundaries for one tract that was classified as a 20-mile/32.19-km food desert and a transit-dependent or 20-mile/32.19-km food desert.

#### 2.2.3. Social Environment: Socioeconomic Status and Racial Similarity

We operationalized the social environments of schools with two different ratios that approximate the extent of demographic similarity between schools and their neighborhoods. In each measure, we used the 2014–2015 school-level data from the U.S. Department of Education, National Center for Education Statistics, Common Core of Data, as the numerator [61]. We operationalized school SES by drawing on the common indicator of the percentage of students who are eligible for the NSLP [37]. Three high schools were missing data on students eligible for NSLP. In two of the cases, we used High-schools.com as an alternative source for accessing National Center for Education Statistics high school-level NSLP data from 2015–2016 [62]. In the third case, High-schools.com was also the source, but the percentage of students eligible for the NSLP at the closest elementary school to the high school was used as a proxy because high school-level NSLP was still not reported. We measure the racial composition of schools with the percent of students who identify as “non-Hispanic white,” to which we refer for brevity as “white”.

Neighborhoods serve as the denominator in our social environment ratios. We constructed neighborhoods with 2010 Decennial Census block-level data because blocks permit more precise measures than other census geographies of the demographic composition of the 500-m, Euclidean, pedestrian environment around schools [59]. Figure 2 illustrates how we measured neighborhoods around the schools and constructed our social environment variables with two example schools that achieved the closest degree of similarity with regard to SES (Willamette High School in Eugene, Oregon) and white composition (Colton High School in Colton, Oregon). As shown in the figure, we first identified which blocks intersected each school’s 500-m Euclidean buffer. We then summed each block-level SES and racial counts and divided them by their respective universes of housing units and population to create aggregated, neighborhood-level percent measures. We use the percent of renter-occupied housing units as our neighborhood-level denominator for the SES environment variable for two reasons. First, housing tenure is a dimension of U.S. social stratification. That is, homeownership reflects upward social mobility and access to channels of wealth accumulation while renting represents lower SES and relatively limited opportunities for wealth accumulation in the housing market [63,64,65]. Second, no other comparable SES measures exist at the census block level. We matched our school-level racial status indicator with the percent of white residents in the neighborhood.

From these preliminary school- and neighborhood-level SES and racial indicators, we calculated our school-to-neighborhood social environment ratio variables. We operationalized SES similarity with the ratio of the *percent of students eligible for NSLP to the percent of neighborhood renter-occupied housing units* (see Figure 2A). We operationalized racial similarity with the ratio of the *percent of white students to the percent of neighborhood white population* (see Figure 2B). Values equal to one for each ratio reflect the similarity between the schools and their neighborhood composition. Values greater than one for the SES and racial similarity ratios indicate, respectively, higher school-level concentrations of low SES students than neighborhood SES and higher school-level white composition than neighborhood white composition. Values less than one for each ratio represent the inverse of these social environment relationships.

#### 2.2.4. Organizational Environment: Neighboring Open Campuses

We operationalized the organizational environment of schools with the *percent of the nearest 45 high schools that had open campus policies*. Our guiding theoretical considerations informed the development of this variable [27,28,29,41,42,43,44,45,46,47]. In constructing this variable, we also sought to address spatial dependence in the logistic regression model. We defined neighboring schools with a Euclidean-based 45-nearest-neighbor-threshold after finding that this neighborhood threshold enabled us to successfully address spatial autocorrelation in the residuals from the logistic regression analysis.

### 2.3. Regression Methodology

We used binary logistic regression analyses to model open campus policy holding as a function of built, food, social, and organizational environment influences. Each model used the following equation in IBM SPSS Statistics for Windows, Version 25.0. (Armonk, NY, USA) [66]:(1)$$\log\left(\frac{P}{1-P}\right)=\alpha +\sum \beta_{k}X_{k},$$
where log(*P*/1 − *P*) was the natural log of the *P* probability of an Oregon public high school holding an open campus policy in 2017, α was the intercept, and $\sum \beta_{k}X_{k}$ is the sum of β coefficients for the *k* number of *X* independent variables [66,67].

To maintain consistency with the 45-nearest-neighbor threshold used to measure the spatial extent of the influential organizational environment on schools in this study, we used the same 45-nearest-neighbor spatial weights matrix to diagnose the degree of spatial dependence in our regression models. This spatial weights matrix is row-standardized, and it resulted in 22.50% spatial connectivity between schools.

## 3. Results

### 3.1. Descriptive Statistics

Table 1 provides descriptive statistics for each study variable. The majority (*n* = 134, 67%) of the 200 Oregon high schools in this sample had open campus policies. The mean walkability score (35.54) for the built environments surrounding the schools indicates low walkability (i.e., “most errands require a car”) [51]. The large standard deviation (*SD* = 23.19) around the mean reflects wide variability in walkability across school settings. The large majority of the 200 schools had walkability scores within the lowest three of the five walkability score ranges: 71 (35.50%) in which “almost all errands require a car”, 71 (35.50%) where “most errands require a car”, and 43 (21.50%) in which “some errands can be accomplished on foot”.

Of the four different food environment variables, we see that schools, on average, had more of their pedestrian environment covered by food desert tracts that are one mile from a supermarket. However, that particular measure of food desert tracts has the greatest amount of variability (*SD* = 41.97%). On average, almost a quarter of the schools’ pedestrian environment was exposed to a food desert tract that is transit-dependent or 20 miles from a supermarket, and there is less variability in exposure to such food desert tracts than there is to one-mile food desert tracts.

There are other noteworthy characteristics of our sample. Regarding the social environment, the average SES of the schools tended to be lower than its surrounding neighborhood, as indicated by the higher concentration of low-income students than neighborhood renters. The racial makeup of the schools also did not tend to match the racial makeup of the schools’ surrounding neighborhood. On average, the schools had lower concentrations of white students than there were white residents in the surrounding neighborhood. Finally, a mean of 63.54% of the closest 45 high schools in the sample had an open campus policy.

### 3.2. Average Differences between Open and Closed Campus Schools

Table 2 compares the independent variable means by open campus policy holding. There are some noteworthy patterns evident in the table regarding environmental differences between open campus and closed campus schools. Schools with open campus policies had higher mean walkability scores than closed campus schools. However, open campus schools had less mean exposure to 20-mile food desert tracts and greater mean exposure to transit-dependent or 20-mile food deserts than closed campus schools. With regard to the social environment variables, open campus schools had slightly higher ratios of white students relative to neighborhood white composition than closed campus schools. In contrast, closed campus schools had higher ratios of low-income students relative to neighborhood renters than open campus schools. Lastly, open campus schools had greater average shares of their neighboring schools that also hold open campus policies than schools with closed campus policies.

### 3.3. Logistic Regression Analyses

Table 3 summarizes the results from our logistic regression analyses. Odds ratios greater than one suggest a one-unit increase in the independent variable is associated with a greater likelihood of a school having an open campus policy. Odds ratios less than one indicate a one-unit increase in the independent variable is associated with decreased odds of a school having an open campus policy. Both models presented in Table 3 have statistically significant model chi-square values, which suggest the models significantly and reliably predict a school’s open campus policy.

Model 1 establishes the effects of the built, food, and social environment variables on a school’s open campus policy holding. In this multivariable context, we see that only three variables are significant predictors of open campus policy holding. Specifically, a one-point increase in walkability is associated with a 2.8% increase in the likelihood of a school having an open campus policy (odds ratio (OR) = 1.028; 95% confidence interval (CI) = 1.011 to 1.045). In contrast, a one-point increase in the percent of a school’s 500-m buffer covered by a 20-mile food desert is associated with a 2.1% decrease in the odds of a school holding an open campus policy (OR = 0.979; 95% CI = 0.959 to 0.999). Lastly, only the racial component of the school’s social environment is significant in Model 1, and it exhibits a very large effect in that model. That is, a one-point increase in the ratio of the percent of white school students to the percent of neighborhood white composition is associated with a 735% increase in the odds of a tract maintaining an open campus policy (OR = 8.348; 95% CI = 1.001 to 69.647).

Figure 3 shows the Moran’s I values for the residuals from Model 1, which excludes the organizational environment variable. Moran’s I is a correlation coefficient that measures the degree of autocorrelation within attribute values of a variable of interest across geographic space. Moran’s I values near zero indicate a random distribution while those closer to one indicate complete clustering. As shown in Figure 3, the Moran’s I of the residuals exhibited a minor peak in value from 40 to 45 nearest neighbors, followed by a decline in value from 45 to 50 nearest neighbors. Given these patterns, we explored different effects using those nearest neighbor thresholds and found that the 45 nearest neighbor threshold adequately characterized the spatial extent of the organizational environment of Oregon public high schools that influences whether a focal Oregon public high school held an open campus policy. We incorporated these insights in the construction of our organizational environment variable, which we add to Model 2.

In Model 2, we see that a one-percentage point increase in the nearest 45 high schools that have an open campus policy is significantly associated with a 6.2% increase in the odds of the focal school within that organizational environment having an open campus policy (OR = 1.062; 95% CI = 1.036 to 1.088). Upon including this organizational environment factor, the significant effect of the racial aspect of the social environment on the school’s open campus policy holding that was evident in Model 1 goes away (Model 2; OR = 4.801; 95% CI = 0.490 to 47.015). However, the effects of the walkability (OR = 1.035; 95% CI = 1.017 to 1.054) and the 20-mile food desert (OR = 0.977; 95% CI = 0.957 to 0.998) variables maintain their magnitude and statistical significance with the inclusion of the organizational environment variable in Model 2.

Including the organizational environment variable in Model 2 also better accounts for spatial dependence in the predictors of open campus policy holding—as seen in the comparison of the Moran’s I tests of the model regression residuals in both models. Without the organizational variable in Model 1, the regression residuals in that model exhibit significant spatial autocorrelation. Including the organizational environment variable eliminates the spatial dependence in Model 2 and contributes to better model fit with a higher pseudo R^2^ value and smaller −2 log likelihood value [61].

## 4. Discussion

In this study, we described the prevalence of open campus policies among Oregon public high schools, and identified multiple environmental predictors of open campus policy holding. We found a high prevalence (67%) of open campus policies across Oregon public high schools, which is greater than the national norm. Two national reports on school policies, published in 2007 and 2016, indicate that, respectively, only 25% and 37.1% of U.S. high schools have open campus policies [68,69]. These findings could reflect real differences in the prevalence of open campus policies in Oregon compared with other states, lending credence for further studying the Oregon context related to open campus policies and their implications for students’ health. The nine years between the two national reports and the +12% difference in open campus policy prevalence could reflect a national trend toward embracing open campus policies among U.S. public high schools. This would be surprising considering open campus policies have raised safety concerns and student safety is a salient issue in the United States [14]. The United States has seen a high incidence of fatalities from school shootings, primarily taking place in high schools, between 2005 and 2018 [70]. This would be reason for schools to tighten their control over the comings and goings of students through closed campus policies, rather than increase their openness. Surveillance from the Centers for Disease Control and Prevention even indicates that lunch time is a common time for school-associated violent deaths [71].

Further, the higher prevalence of open campus policies compared with other states may put Oregon high school students at an increased risk for poor dietary intake during the school year. Within Oregon, students attending schools with open campus policies are exposed to a greater number of food options, which is associated with greater fast food consumption and lower participation in NSLP, and poorer dietary intake among NSLP-qualified students [6,13,16]. However, these are associations, and there is a need to evaluate the impact of open campus policies on students’ lunch time food purchasing, dietary, and physical activity behaviors across rural and urban settings with varying levels of walkability and access to healthy and affordable food. Within the School Nutrition Dietary Report III, all principals of schools with open campus policies reported that there was at least one food vendor in walking distance from their school and 76% of those food vendors were fast food restaurants [69]. The nutrition standards for NSLP and competitive foods sold on school campuses have improved due to the Healthy, Hunger-Free Kids Act of 2010. However, if high school students continue to be permitted to acquire off campus food, then they will likely continue to eat fast food and other unhealthy foods at higher rates than students who are limited to on-campus foods [72,73]. Closed campus policies could address the exposure to unhealthy foods, and in turn, consumption of unhealthy foods and disparate risk for excess weight gain and associated chronic diseases [74,75].

Our findings suggest that the environments around the schools matter when it comes to whether an Oregon high school has an open or closed campus. Specifically, the likelihood of an Oregon public high school having an open campus policy was most consistently associated with higher walkability scores of the area around the school, less exposure to 20-mile food desert tracts, and higher shares of neighboring schools with open campus policies.

Related to the built environment, we found Oregon public high schools in areas with greater walkability were more likely to have open campus policies. High walkability, as measured in this study using Walk Scores, describes high population and traffic density, as well as highly mixed residential and commercial land uses that allow for at least some errands to be accomplished without a car [52]. While Walk Scores do not assess pedestrian-friendly infrastructure (e.g., sidewalks and crosswalks), Walk Scores are positively associated with these types of infrastructure [54]. Overall safety is also not included in Walk Score calculations. Perceived safety is a factor within a well-validated measure of neighborhood walkability that is positively associated with self-reported built environment characteristics, like sidewalks and amenities within walking distance [76,77]. A rarer study using objective measures of safety and walkability found that objective measures of crime incidence were not associated with Walk Scores [78]. Our study’s finding suggests that district and school policy decision-makers may take into account the walkability around the school when considering open campus policies. How easily high school students can travel to amenities, particularly food vendors, without having access to a car is a pragmatic consideration related to open campus policy holding.

The relationship between walkability and schools’ open campus policy holding is consistent with our findings regarding the effects of different indicators of schools’ food environment on their open campus policy holding. That is, we find that open-campus schools had less exposure, on average, to 20-mile food desert tracts. All eight schools that were exposed to 20-mile food desert tracts in their 500-m pedestrian context were located in rural settings of central and eastern Oregon. Of those eight, four were open campuses with an average walkability score of 24.25 while the other four were closed campuses with a lower average walkability score of 18. When contextualized this way and in relation to the positive association between walkability and open campus policy holding, our results suggest that Oregon public high schools’ level of walkability is an important predictor of those schools’ open campus policy holding in rural and urban food environments. Substantively, these results suggest that schools with relatively high walkability scores and some access to a supermarket could contribute to the decision to adopt an open campus policy. Similarly, if schools have low walkability and nowhere to access healthy foods for lunch in their surrounding food environment, then there may be no reason to permit them to leave campus during lunch, thus contributing to a decision to have a closed campus.

Our findings also support the notion that the organizational environment influences Oregon public high schools’ open campus policy holding. We found that the proportion of the closest 45 high schools with open campus policies had an independent and positive effect on the likelihood of open campus policy holding. These findings align with our guiding ecological, systems, organizational, and institutional theoretical frameworks [27,28,29,41,42,43,44,45,46,47]. In particular, they suggest that dominant norms, structures, and practices within an organizational field, such as holding an open campus policy, can be diffuse throughout that field and influence how similar organizations (in this case, public high schools) act in a coordinated manner in time and space [27,28,29]. As illustrated in Figure 4, this dynamic is not fully explained by school district policy. Rather, it appears to be consistently related to the spatial context in which Oregon public high schools operate.

The 200 public high schools that we analyzed reside within 146 school districts. Open campus policy holding was uniform across public high schools within 143 of those school districts. This consistency within districts is likely in part because Oregon school districts, as opposed to the state, have most of the governing power to decide local policies [79]. However, five of the six (83%) public high schools in the Salem–Keizer 24J School District (see Figure 4) held an open campus policy. Meanwhile, three of the six (50%) public high schools in the Klamath County School District held an open campus policy, and two of the three (67%) public high schools in the Gresham–Barlow 10J School District held an open campus policy. As our statistical analyses and Figure 4 suggest, open campus and closed campus schools tend to be proximate to one another and consistently so within their nearest 45-school organizational environment.

We considered the SES and racial components of schools’ social environment because of the long history of racial exclusion and school and residential segregation that have benefited white and higher SES people in the United States [36,37] and, especially, in Oregon [38,39,40]. We found that Oregon public high schools, regardless of their open-campus policy status, were on average less white than their surrounding neighborhoods, but open campus policy schools were comparatively slightly whiter than their surrounding neighborhoods. These patterns appear to coincide with the finding that U.S. public schools have, on average, higher nonwhite composition [37] than their surrounding neighborhoods, which may be attributed to the flight of white families from public schools. The effect of our racial similarity variable—i.e., the ratio of the percent of white students to neighborhood white population—stood out as a very powerful and positive predictor of open campus policy holding in our first regression model. We did not find the same initial results for our SES social environment variable. However, the introduction of the organizational environment variable into Model 2 explained away the effects of the white racial environment variable on open campus policy holding. These results may be attributed to the marginal differences between open- and closed-campus high schools’ racial similarity ratios. These results may also be influenced by the fact that Oregon’s highly white (and “non-Hispanic”) demographic composition (75.3%) may not introduce enough variability in these racial ratios (i.e., majority white in schools and the surrounding areas) to detect significant effects on campus policy holding, even if they were to exist [80]. However, there was SES variability, and these ratios were not significantly associated with open campus policy holding after other environmental factors were added in the regression analyses. Future research should examine these social environmental factors and their associations with open campus policies in states with more heterogeneous racial demographics to see how they fare compared with other environmental factors.

A limitation of this study is that data used in these analyses were collected across different years, ranging from 2010 to 2017. It is possible that our 2010 Decennial Census data did not accurately reflect schools’ environments in 2017 when open campus policy holding data were collected. Additionally, aggregate data were used, thus potential individual level confounders were unmeasured and inferences based on these study findings are limited to the aggregate level. Furthermore, at the time of this study, we did not possess high-resolution, geocoded food outlets data that are typically used for mapping food deserts [57]. Future research could adapt our exploratory model by replacing the coarse food environment variables that we developed from previous research [58] with alternative indicators of food deserts and insecurity. These alternatives could be based on a more comprehensive understanding of the food environment (i.e., beyond supermarkets) and primary data collection and analysis techniques [57,81,82,83]. To the authors’ knowledge, this is the first study that identifies factors associated with open campus policy holding. However, we can infer that there are economic, political, and other unknown reasons not included in our study that also contribute to open campus policy holding. This is an additional area for future research. Furthermore, the implications for open campus policies would surely vary across countries as cultural norms (e.g., going home for lunch) and other school policies (e.g., longer lunch periods) differ across countries, and more research is needed on the topic.

There are important, practical implications of our findings. These relate to evidence-based decision-making in the fields of education and public health, which outline the importance of using the best available prior evidence to maximize positive impacts from an intervention or policy decision [84,85,86]. Findings from this study provide helpful insight into the multiple environmental factors associated with open campus policy holding that were previously absent from the literature and potentially even unknown to the policy decision-makers themselves. Building an Oregon-specific, empirical understanding of these relevant factors associated with open campus policies and simultaneously raising awareness of the behavioral and health outcomes associated with open campus policies, may prompt Oregon school district and high school administrators to reconsider their open campus policy holding. Or, perhaps, our study may reaffirm their policy decision with more clarity of the multiple environmental influences on school policy.

## 5. Conclusions

This study contributes to previous research by advancing a new empirical analysis of the prevalence of open campus policies among Oregon public high schools, and it is the first to identify multiple environmental predictors of open campus policy holding. The findings indicate that the majority of Oregon public high schools had open campus policies as of 2017. Further, of the factors considered in our analysis, the likelihood of an Oregon public high school having an open campus policy was most consistently predicted by schools’ built-, food-, and organizational-environmental influences. These findings provide an empirical understanding of the associated multiple environmental predictors of open campus policy holding, and they enable more informed decisions moving forward regarding a policy with potential health implications for Oregon students.

## Figures and Tables

**Figure 1 ijerph-17-00469-f001:**
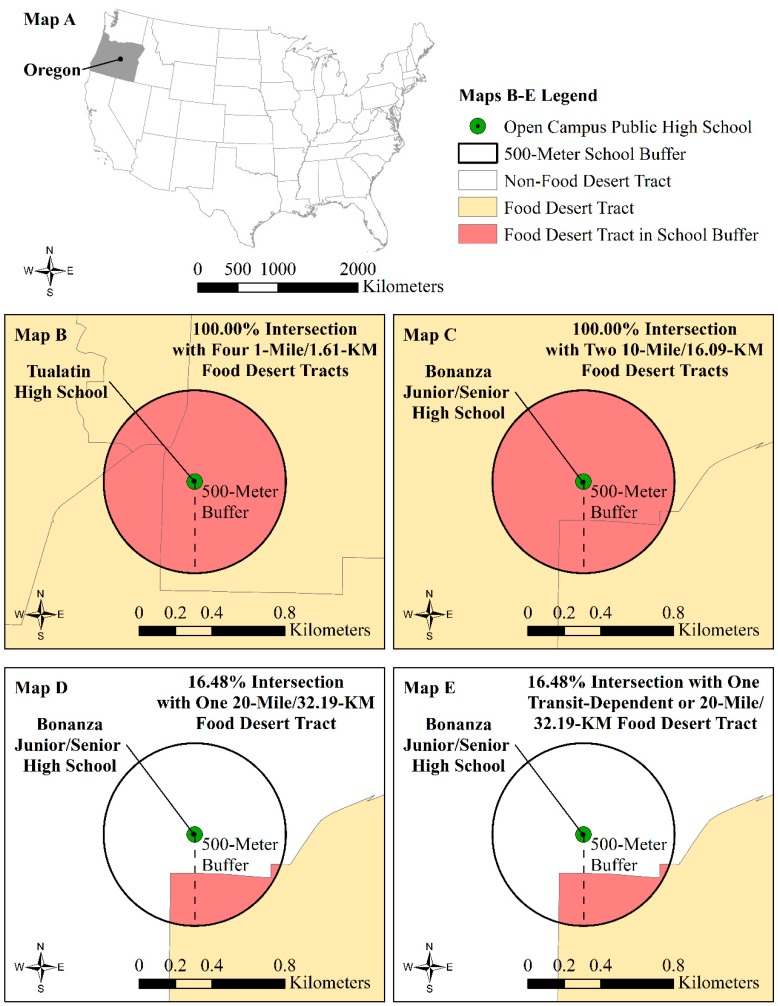
(**A**) Oregon and continental United States boundaries; (**B**–**E**) illustration of food desert tract intersection with a 500-m buffer for Oregon public schools to derive Oregon public high schools’ four food environment (food desert exposure) variables.

**Figure 2 ijerph-17-00469-f002:**
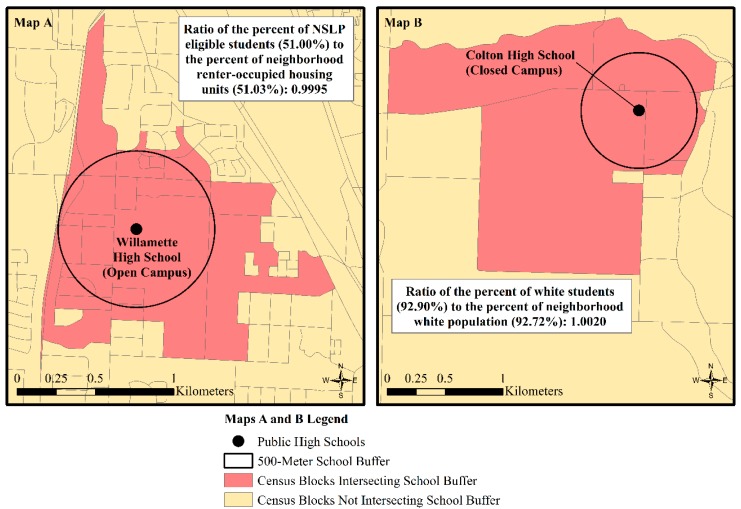
Illustration of census block intersection with a 500-m buffer for Oregon public high schools to derive Oregon public high schools’ social environment variables of (**A**) socioeconomic status similarity and (**B**) racial similarity.

**Figure 3 ijerph-17-00469-f003:**
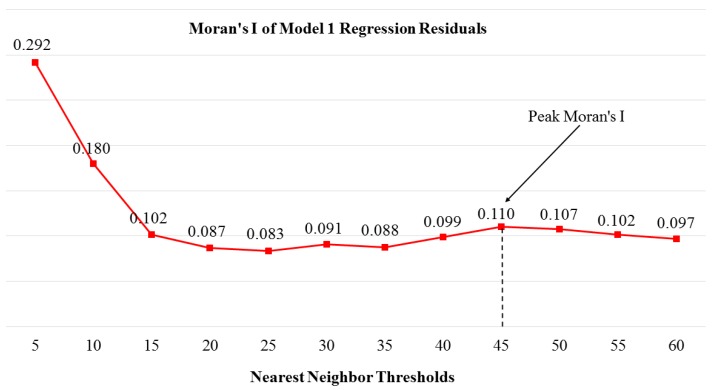
Results from the spatial autocorrelation analysis of the residuals from the logistic regression model without the organizational environment variable (see Model 1, Table 3). The peak Moran’s I value of 0.110 at the 45 nearest neighbor threshold motivated the use of the 45 nearest neighbor threshold for the organizational environment independent variable in the full regression model (Model 2, Table 3).

**Figure 4 ijerph-17-00469-f004:**
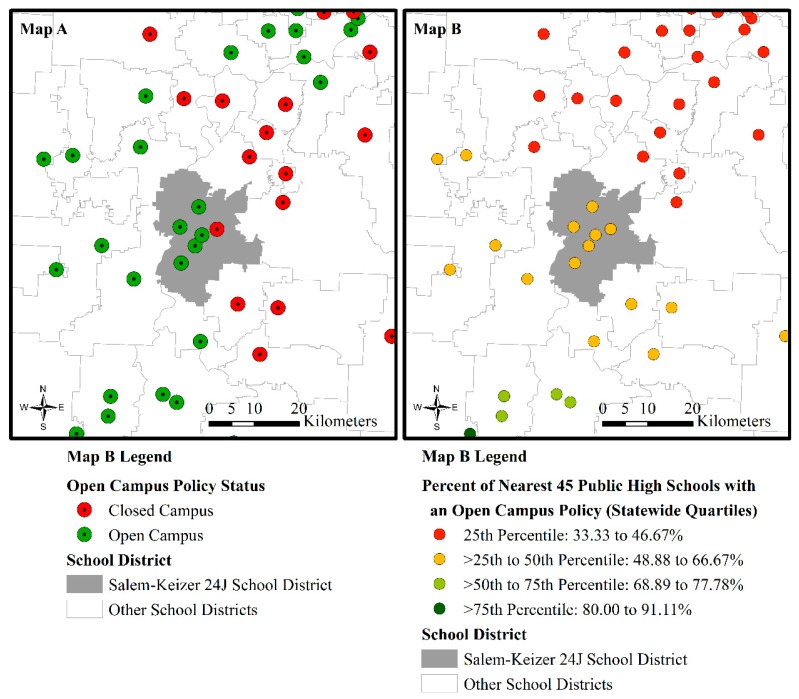
(**A**) Open campus policy status of Oregon public high schools and (**B**) the percent of nearest 45 public high schools in relation to the example case of the Salem–Keizer 24J School District and neighboring school districts.

**Table 1 ijerph-17-00469-t001:** Descriptive statistics for independent variables used in the logistic regression analysis (*n* = 200 public high schools).

Variables	Mean	*SD*	Minimum	Maximum
*Dependent variable*				
School has open campus policy	0.67	0.47	0.00	1.00
*Built environment*				
Walkability	35.54	23.19	0.00	95.00
*Food environment*				
Percent of school 500-m buffer covered by food desert tract:				
1 miles from a supermarket	28.43	41.97	0.00	100.00
10 miles from a supermarket	14.37	34.77	0.00	100.00
20 miles from a supermarket	3.57	18.40	0.00	100.00
Transit-dependent or 20 miles from a supermarket	23.10	38.67	0.00	100.00
*Social environment*				
Ratio of school-to-neighborhood demographic composition:				
Percent of National School Lunch Program eligible students to the percent of neighborhood renter-occupied housing units	1.96	1.95	0.00	24.27
Percent of white students to the percent of neighborhood white population	0.85	0.15	0.19	1.23
*Organizational environment*				
Percent of nearest 45 high schools that have open campus policies	63.54	15.72	33.33	91.11

**Table 2 ijerph-17-00469-t002:** Comparison of independent variable means by open campus policy holding (*n* = 200 public high schools).

Variables	Open Campus	Closed Campus
*Built environment*		
Walkability	39.31	27.88
*Food environment*		
Percent of school 500-m buffer covered by food desert tract:		
1 miles from a supermarket	28.42	28.44
10 miles from a supermarket	14.73	13.64
20 miles from a supermarket	2.36	6.03
Transit-dependent or 20 miles from a supermarket	25.28	18.69
*Social environment*		
Ratio of school-to-neighborhood demographic composition:		
Percent of National School Lunch Program eligible students to the percent of neighborhood renter-occupied housing units	1.85	2.17
Ratio of the percent of white students to the percent of neighborhood white population	0.86	0.83
*Organizational environment*		
Percent of nearest 45 high schools that have open campus policies	67.16	56.20
*n*	134	66

**Table 3 ijerph-17-00469-t003:** Logistic regression results for open campus policy holding (*n* = 200 public high schools).

Variables	Model 1		Model 2	
	Odds Ratio ^1^	95% Confidence Interval	Odds Ratio	95% Confidence Interval
*Built environment*				
Walkability	1.028 ***	1.011 to 1.045	1.035 ***	1.017 to 1.054
*Food environment*				
Percent of school 500-m buffer covered by food desert tract:				
1 miles from a supermarket	1.000	0.992 to 1.008	1.002	0.993 to 1.011
10 miles from a supermarket	1.007	0.996 to 1.019	1.001	0.989 to 1.014
20 miles from a supermarket	0.979 *	0.959 to 0.999	0.977 *	0.957 to 0.998
Transit-dependent or 20 miles from a supermarket	1.008	0.998 to 1.017	1.007	0.997 to 1.018
*Social environment*				
Ratio of school-to-neighborhood demographic composition:				
Percent of National School Lunch Program eligible students to the percent of neighborhood renter-occupied housing units	1.037	0.882 to 1.220	1.049	0.891 to 1.234
Percent of white students to the percent of neighborhood white population	8.348 *	1.001 to 69.647	4.801	0.490 to 47.015
*Organizational environment*				
Percent of nearest 45 high schools that have open campus policies			1.062 ***	1.036 to 1.088
*Model diagnostics*				
−2 Log likelihood	232.653		205.121	
Model chi-square	21.019 **		48.551 ***	
Degrees of freedom	7		8	
Pseudo R-squared	0.139		0.300	
Moran’s I for regression residuals ^2^	0.110 ***		−0.008	

^1^ Odds ratios for the constant not displayed in the table; ^2^ Moran’s I test of residuals conducted with 9999 permutations and a 45-nearest neighbor spatial weights matrix. * Correlation is significant at the 0.05 level (two-tailed). ** Correlation is significant at the 0.01 level (two-tailed). *** Correlation is significant at the 0.001 level (two-tailed).

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
