# Peer review of "Open Campus Policies: How Built, Food, Social, and Organizational Environments Matter for Oregon’s Public High School Students’ Health"

_ijerph, 2020, doi:10.3390/ijerph17020469_

Round 1
Reviewer 1 Report
Thank you for the opportunity to review an interesting paper that contributes to the published literature and what is known regarding food access.
The abstract is an excellent overview of headline results but greater insight into the nature of the results could usefully be added.
The introduction and background are logically structured and signpost the reader well to the expected content and analysis.
The methods section is well explained. Line 207: articulate the nature and method of the alternative sources rather than merely signpost the reader to it.
The results section is well articulated and statistically compared.
Further research into students availing of open campus policies freedoms and their food purchasing / consumption behaviors would be worthwhile and complement these data - your recommendation for this as further research is to be welcomed in Lines 470 - 474.
I note, in your conclusion, you cite the merits of your study informing policy decisions re: schools' decisions to operate open or closed campus policies; however, I believe you could arrive at an evidence-informed summative conclusion about whether high schools should have a greater uptake of closed campus policies for health (and also economic and security) reasons.
Miscellaneous:
Line 76: Delete unnecessary extra space before 'school policy-makers'
Line 96: The sentence seems incomplete. Should it read: "valued ... more(?) than the rights of black students"
Line 457: Replace 'Although,' with 'However,'
Line 460: Replace 'fair' with 'fare'
Line 491: Delete unnecessary extra space before the 'likelihood'
Author Response
We thank the reviewers for their thoughtful feedback. We appreciate the opportunity to address the feedback and provide revisions. Our response to each reviewer comment in appears below.
Reviewer 1’s comments and authors’ responses:
Comment 1: The methods section is well explained. Line 207: articulate the nature and method of the alternative sources rather than merely signpost the reader to it.
Response: Pg. 6, Lns 24-28 now read: “In two of the cases, we used High-schools.com as an alternative source for accessing National Center for Education Statistics high school-level NSLP data from 2015-2016 [61]. In the third case, High-schools.com was also the source, but the percentage of students eligible for the NSLP at the closest elementary school to the high school was used as a proxy because high school-level NSLP was still not reported.”
Comment 2: Further research into students availing of open campus policies freedoms and their food purchasing / consumption behaviors would be worthwhile and complement these data - your recommendation for this as further research is to be welcomed in Lines 470 - 474.
Response: “Food purchasing” was added to the recommendation on Pg. 6, Lns 402-402. It now reads, “However, these are associations, and there is a need to evaluate the impact of open campus policies on students’ lunch time food purchasing, dietary, and physical activity behaviors across rural and urban settings with varying levels of walkability and access to healthy and affordable food.”
Comment 3: I note, in your conclusion, you cite the merits of your study informing policy decisions re: schools' decisions to operate open or closed campus policies; however, I believe you could arrive at an evidence-informed summative conclusion about whether high schools should have a greater uptake of closed campus policies for health (and also economic and security) reasons.
Response: We appreciate this comment from the reviewer, but we are afraid it slightly misrepresents our statements in the conclusion section of the original submission. In that original conclusion on Pg. 15, Lns 532-534, we stated that the merits of the study were that it “contributed to previous research by advancing a new empirical analysis of the prevalence of open campus policies among Oregon public high schools, and it is the first to identify multiple environmental predictors of open campus policy holding.” In the original submission and in the revised submission, we make no claims about whether or not Oregon public high schools should have open or closed campus policies because our data do not speak to such issues.
Comment 4: Line 76: Delete unnecessary extra space before 'school policy-makers'
Response: The extra space was deleted on Pg. 2, Ln 84.
Comment 5: Line 96: The sentence seems incomplete. Should it read: "valued ... more(?) than the rights of black students"
Response: We appreciate the reviewer catching this error. That portion of the quote was indeed missing “more.” It has now been corrected to: “valued the desires of white parents more than the rights of black students” on Pg. 3, Ln 104.
Comment 6: Line 457: Replace 'Although,' with 'However,'
Response: “Although” was replaced with “however” on Pg. 14, Ln 193.
Comment 7: Line 460: Replace 'fair' with 'fare'
Response: “Fair” was replaced with “fare” on Pg. 14, Ln 497.
Comment 8: Line 491: Delete unnecessary extra space before the 'likelihood'
Response: The extra space was deleted on Pg. 15, Ln 536.
Reviewer 2 Report
In line 42, do you mean convenience?
As this is my area of research, I find your background to be a bit lacking in the nuances associated with adolescent obesity and eating practices. Particularly, the first paragraph seems to be three statements back to back about associations, but there are many factors that may be associated with poor dietary habits in youth. I think the authors must address this in a concise way.
The authors need to define what open campus policy really means for their research. My understanding is that there are additional hierarchies within that term as well.
There are some additional nuances at play about determining open campus policies as well. I would argue that there needs to be some discussion about how open campus policies are determined primarily.
Aren't national averages of participation in NSLP in the US fairly low for high school students, regardless of open or closed campus policies? I would address this but compare it as being lower (how significantly so in the manuscript).
I would say that your case study sampling was purposeful, which is typical in case study research. You justify the need well, but you do not address the actual sampling approach.
I do appreciate your discussion concerning the specific context of Oregon's environment in order to better understand the research.
I completely agree with the stated limitation regarding not addressing student health behaviors. Although it was outside of the scope of the research question, I think the assumption in the discussion that closed campus policies may decrease poor dietary behaviors, thus resulting in better health outcomes is fairly weak. We know that adolescents' dietary quality is poor, and I might argue that unhealthy food items may still be chosen with a closed campus.
I do think this overall research is critical to the food environment field. We are needing some policy analysis to assist with decision making regarding the higher levels of the socioecological model and behavior change with adolescents. And this research does assist with that.
Author Response
We thank the reviewers for their thoughtful feedback. We appreciate the opportunity to address the feedback and provide revisions. Our response to each reviewer comment in appears below.
Reviewer 2’s comments and authors’ responses:
Comment 1: In line 42, do you mean convenience?
Response: “Convenient” was replaced with “convenience” on Pg. 2, Ln. 47.
Comment 2: As this is my area of research, I find your background to be a bit lacking in the nuances associated with adolescent obesity and eating practices. Particularly, the first paragraph seems to be three statements back to back about associations, but there are many factors that may be associated with poor dietary habits in youth. I think the authors must address this in a concise way.
Response: We have added the following sentence and associated citations to clarify in a very concise way that we recognize that adolescent eating practices are influenced by many factors beyond the environmental ones discussed in this study.
Pg. 1, Lns 34-39 now reads, “An extensive body of research attributes youth’s dietary behavior to varied cognitive, affective, habitual, and environmental influences [2]. Empirical research is increasingly concerned with assessing the environmental influences on youth’s diets [2], reinforcing the notion that environmental-level interventions hold the most promise for producing population level changes [3]. Systematic reviews [4,5] have indicated that future research should pay better attention to schools and their food policies that can affect youth dietary behaviors.”
Comment 3: The authors need to define what open campus policy really means for their research. My understanding is that there are additional hierarchies within that term as well.
Response: The following details were added to Pg. 3, Lns 133-139: “Campus policies were originally organized into three categories: closed campus (students are not allowed to leave campus at any time during the school day); open campus (students are allowed to leave campus at lunch and during periods they do not have class); and restricted open campus (some students are permitted to leave at lunch or the school has additional provisions in place to supervise and regulate open lunch periods). For the purposes of these analyses, open campus and restricted open campus policies were collapsed into open campus policy holding, consistent with other studies [14].”
Comment 4: There are some additional nuances at play about determining open campus policies as well. I would argue that there needs to be some discussion about how open campus policies are determined primarily.
Response: The following was added to the Limitations paragraph on Pgs. 14-15, Lns 513-516: “To the authors’ knowledge, this is the first study that identifies factors associated with open campus policy holding. However, we can infer that there are economic, political, and other unknown reasons not included in our study that also contribute to policy holding. This is an additional area for future research.”
Comment 5: Aren't national averages of participation in NSLP in the US fairly low for high school students, regardless of open or closed campus policies? I would address this but compare it as being lower (how significantly so in the manuscript).
Response: The following was added to the Introduction on Pg. 2, Lns 61-64: “In a study of U.S. schools across grade levels, approximately 40% of students eligible for the NSLP participate, and rates of NSLP participation among high school students averages about half of that of elementary school students [15].”
Comment 6: I would say that your case study sampling was purposeful, which is typical in case study research. You justify the need well, but you do not address the actual sampling approach.
Response: The sampling approach was not further specified because all of the public non-alternative high schools in Oregon were included, rather than a sample of them.
Comment 7: I completely agree with the stated limitation regarding not addressing student health behaviors. Although it was outside of the scope of the research question, I think the assumption in the discussion that closed campus policies may decrease poor dietary behaviors, thus resulting in better health outcomes is fairly weak. We know that adolescents' dietary quality is poor, and I might argue that unhealthy food items may still be chosen with a closed campus.
Response: We appreciate this comment from the reviewer We addressed it in a number of ways. First, we acknowledge this point about the potential exposure to unhealthy foods for adolescents in open and closed campus settings in the first sentence of the abstract, which now states: “Open campus policies that grant access to the off-campus food environment may influence U.S. high school students’ exposure to unhealthy foods, yet predictors of these policies are unknown” on Pg. 1, Lns 14-16.
The following was moved from Limitations to the Discussion on Pg. 12, Lns 402-404: “However, these are associations, and there is a need to evaluate the impact of open campus policies on students’ lunch time food purchasing, dietary, and physical activity behaviors across rural and urban settings with varying levels of walkability and access to healthy and affordable food.”
Additionally, the word, “potential” was added to the last sentence of the Conclusions to acknowledge that more research is needed on how such a policy impacts youth’s dietary behaviors. It now reads, “These findings provide an empirical understanding of the associated multiple environmental predictors of open campus policy holding, and enable more informed decisions moving forward regarding a policy with potential health implications for Oregon students.” Pg. 15, Lns 538-540.
Reviewer 3 Report
This is a well-written manuscript that touches a timely topic. I do only have a few minor comments:
The existence of a ‘research gap’ is clearly emphasized; however, the authors could further elaborate on how this study adds value to the international literature. Can you provide more details on why you decided to measure food deserts, instead of measuring youth accessibility to unhealthy foods off-campus? Please make it clear whether you used Euclidian buffers and Euclidian distances or street network distances. I would suggest to include confidence intervals in the text, in order to provide more insights about the magnitude of the associations found by the authors. This study is with characteristics of ecological study, as the information was not individually collected. Please state the limitations of this sort of study design.Author Response
We thank the reviewers for their thoughtful feedback. We appreciate the opportunity to address the feedback and provide revisions. Our response to each reviewer comment in appears below.
Reviewer 3’s comments and authors’ responses:
Comment 1: The existence of a ‘research gap’ is clearly emphasized; however, the authors could further elaborate on how this study adds value to the international literature.
Response: We very briefly elaborated on implications for our study’s findings to the international literature on Pg. 15, Lns 516-519, “Furthermore, the implications for open campus policies would surely vary across countries as cultural norms (e.g., going home for lunch) and other school policies (e.g., longer lunch periods) differ across countries, and more research is needed on the topic.”
Comment 2: Can you provide more details on why you decided to measure food deserts, instead of measuring youth accessibility to unhealthy foods off-campus?
Response: On Pg. 4, Lns 165-169 in Section 2.2.2 we now include the following passage that justifies our use of the U.S.D.A. data on the location of food desert tracts in the United States: “At the time of this study, we did not have access to the geographic locations of establishments typically used in previous food desert research as markers of healthy or unhealthy off-campus food options [51]. Instead, we draw on publicly available results of a U.S. Department of Agriculture-sponsored study to assess school exposure to food desert census tracts within a 500-meter Euclidean buffer around the schools [57].”
Comment 3: Please make it clear whether you used Euclidian buffers and Euclidian distances or street network distances.
Response: Section 2.2 on Pgs. 4-7 now includes explicit statements about our Euclidean-based buffers and distance measurements.
Comment 4: I would suggest to include confidence intervals in the text, in order to provide more insights about the magnitude of the associations found by the authors.
Response: Section 3.3, beginning on Pg. 9, Ln 324, now includes confidence intervals in the text to aid in interpreting the magnitude of the associations found in the logistic regression analyses.
Comment 5: This study is with characteristics of ecological study, as the information was not individually collected. Please state the limitations of this sort of study design.
Response: The following was added to the study limitations on Pg. 14, Lns 501-503: “Additionally, aggregate data were used, thus potential individual level confounders were unmeasured and inferences based on these study findings are limited to the aggregate level.”
Round 2
Reviewer 3 Report
The authors have addressed my comments. I think that the manuscript is suitable for publication in its present form.